# Comparison of Short-Term Post-Thymectomy Outcomes by Time-Weighted Dosages of Drug Requirements between Thymoma and Non-Thymoma Myasthenia Gravis Patients

**DOI:** 10.3390/ijerph20043039

**Published:** 2023-02-09

**Authors:** Phattamon Wiboonpong, Suwanna Setthawatcharawanich, Pat Korathanakhun, Thanyalak Amornpojnimman, Nannapat Pruphetkaew, Pensri Chongphattararot, Chutarat Sathirapanya, Pornchai Sathirapanya

**Affiliations:** 1Department of Internal Medicine, Faculty of Medicine, Prince of Songkla University, Songkhla 90110, Thailand; 2Epidemiology Unit, Faculty of Medicine, Prince of Songkla University, Songkhla 90110, Thailand; 3Department of Family and Preventive Medicine, Faculty of Medicine, Prince of Songkla University, Songkhla 90110, Thailand

**Keywords:** myasthenia gravis, thymoma, thymectomy, exacerbation, crisis

## Abstract

(1) Background: Early thymectomy is suggested in all clinically indicated myasthenia gravis (MG) patients. However, short-term clinical response after thymectomy in MG patients has been limitedly described in the literature. This study aimed to compare the 5-year post-thymectomy outcomes between thymoma (Th) and non-thymoma (non-Th) MG patients. (2) Methods: MG patients aged ≥18 years who underwent transsternal thymectomy and had tissue histopathology reports in Songklanagarind Hospital between 2002 and 2020 were enrolled in a retrospective review. The differences in the baseline demographics and clinical characteristics between ThMG and non-Th MG patients were studied. We compared the time-weighted averages (TWAs) of daily required dosages of pyridostigmine, prednisolone or azathioprine to efficiently maintain daily living activities and earnings between the MG patient groups during 5 consecutive years following thymectomy. Post-thymectomy clinical status, exacerbations or crises were followed. Descriptive statistics were used for analysis with statistical significance set at *p* < 0.05. (3) Results: ThMG patients had significantly older ages of onset and shorter times from the MG diagnosis to thymectomy. Male gender was the only significant factor associated with ThMG. TWAs of the daily MG treatment drug dosages required showed no differences between the groups. Additionally, the rates of exacerbations and crises were not different, but decremental trends were shown in both groups after the thymectomies. (4) Conclusions: The daily dosage requirements of MG treatment drugs were not different. There was a trend of decreasing adverse event rates despite no statistically significant differences during the first 5 years after thymectomy in ThMG and non-ThMG patients.

## 1. Introduction

Myasthenia gravis (MG) is a common post-synaptic neuromuscular junction (NMJ) disorder in which fatigability or reversible muscular weakness of the eye lid, extraocular, bulbar or limb muscles is the clinical hallmark. MG is an immunologic disorder in which acetylcholine (AChR) or other receptors on the post-synaptic membrane, such as muscle-specific kinase (MuSK), titin, ryanodine receptors, etc. are degraded by immunological mechanisms such as complement-mediated receptor degradation by circulating antibodies, accelerated endocytosis, or functional blockade of the post-synaptic receptors. Diverse clinical presentations have been described, depending on the specific type of receptors affected [1,2,3,4,5]. Moreover, it has been known that the thymus of MG patients acts as the primary source of auto-antibody production regardless of thymic histopathology, i.e., either thymoma (Th) or non-thymoma (non-Th). Therefore, early thymectomy is advocated, although obvious clinical benefits may require several years after thymectomy [6,7,8,9,10,11,12,13,14]. Additionally, a number of studies have suggested that early thymectomy was superior to conservative treatment in terms of MG clinical remission, regardless of the thymic tissue histopathology [7,8,9,10,15,16,17]. Currently, the assessment of post-thymectomy outcomes as recommended by the Myasthenia Gravis Foundation of America Post-Intervention Status (MGFA-PIS) requires a neuromuscular specialist, and the evaluation is based only on the groups of drugs, i.e., symptomatic drugs, immunosuppressants or the combination of both, prescribed which were able to minimize or get rid of MG symptoms or signs for at least one year. In real practice, post-thymectomy MG patients were followed-up with variable intervals of time depending on the clinical response to the treatments and the fluctuating nature in the clinical severity of MG itself; therefore, the dosage of the drugs prescribed may have been variable from one visit to another visit during the follow-ups. There is no daily average dosage of each drug required by the MG patients included in the evaluation method. The second part of the MGFA-PIS, which evaluates changes in post-thymectomy clinical status (i.e., improved, unchanged or worse), is based on subjective “increased” or “reduced” symptoms of MG, and it requires a Quantitative Myasthenia Gravis (QMG) score in cases of prospective evaluations. The QMG score evaluation process requires a standard spirometer and a neuromuscular specialist. The subjectivity of the evaluation method and requirements of a spirometer and a neuromuscular specialist limit the generalized application of the MGFA-PIS, especially in the outpatient departments (OPDs) of many medical centers where a spirometer and/or a neuromuscular specialist are not available, such as in our center.

The aims of this study were: (a) to compare the baseline clinical characteristics between histopathology-confirmed ThMG and non-ThMG patients, (b) to compare five consecutive years of post-thymectomy outcomes objectively using time-weighted averages (TWAs) of daily dosages of pyridostigmine (a symptomatic drug), prednisolone or azathioprine (immunosuppressants) which were required by the MG patients to effectively maintain their daily living activities and earn their livings (ADLE). The effective outcomes were self-assessed by the patients and discussed between the patients and their physicians before the dosage adjustment. We think this method of co-decision on treatment can accurately represent the individual MG patient requirement of each MG drug for living well. Additionally, (c) we compared the frequencies of exacerbations, crises and deaths due to MG after thymectomy between the two MG patient groups.

## 2. Materials and Methods

### 2.1. Study Population and Setting

This retrospective study enrolled MG patients aged 18 years or more who underwent an extensive transsternal thymectomy, and for whom thymic tissue histopathology reports were available from 2002 to 2020 in Songklanagarind Hospital of Prince of Songkla University. The hospital is a 900-bed tertiary-care and referral center serving the lower southern provinces of Thailand. We studied the post-thymectomy outcomes of the MG patients who were followed-up in our service from 1 up to 5 years.

### 2.2. Data Collection

We retrospectively collected the patients’ demographic data, pre-thymectomy MGFA classification, time from the diagnosis of MG to thymectomy, thymic tissue histopathology reports, and post-thymectomy daily dosage of the symptomatic drug (pyridostigmine), or immunosuppressant(s) (prednisolone and/or azathioprine) required in the ThMG and non-Th MG patients during the follow-ups at the OPD. We used the TWAs of the MG treatment drugs to report the average daily dosage requirements of the MG drugs to maintain the patients’ ADLEs. We reported the TWAs of the individual drug in six-month intervals, i.e., from 1 January to 30 June and from 1 July to 31 December of each year. The TWA of the daily dosage requirement formula was calculated by the summation of the results from the amount of the individual drug used/day (mg/d) multiplied by the number of days during which the dosage was used, and then, the summation was divided by total days in each six-month interval of a year as follows:$$\frac{\mathrm{MG}\;{\mathrm{drug}}_{1.}\;(\mathrm{mg}/d)\;\times \;{\mathrm{days}}_{1}+\mathrm{MG}\;{\mathrm{drug}}_{2.}\;(\mathrm{mg}/d)\;\times \;{\mathrm{days}}_{2}+\ldots }{\mathrm{Days}\;\mathrm{from}\;1\;\mathrm{January}\;\mathrm{to}\;30\;\mathrm{June}\;(\mathrm{or}\;1\;\mathrm{July}\;\mathrm{to}\;31\;\mathrm{December})}$$

The post-thymectomy outcomes in this study were also evaluated by:(1)“MGFA-PIS” outcome classifications as: complete stable remission (CSR), pharmacologic remission (PR) and minimal manifestations (MM 0–3) following standard definitions.(2)“Changes in clinical status”, for which we only studied “exacerbation”, “crisis” and “died of MG”. We did not include “better”, “unchanged” or “worse” because of their subjectivity as mentioned, and they were replaced by the TWAs of MG drug dosages of our study design. We expanded the definitions of exacerbation, crisis and died of MG for this study as follows:Exacerbation: post-thymectomy MG patients with CSR, PR or MM who had recently developed a clinical progression of MG symptoms significantly greater than that described in each category and had required a symptomatic drug or immune-suppressants (prednisolone or azathioprine) ≥50% of the baseline dosage before the exacerbation.Crisis: the condition of emerging respiratory compromise in a previously stable post-thymectomy clinical status patient, and endotracheal intubation with mechanical ventilation was indicated for respiratory support.Died of MG: an MG patient whose cause of death was directly related to the MG itself or MG treatment complications, or death occurred within 30 days after the thymectomy.

### 2.3. Statistical Analysis

The associations between patient demographic data, prethymectomy MGFA classifications, time from diagnosis of MG to thymectomy, prethymectomy requirement of daily dosages of the MG drugs, and the thymic tissue histopathology results reporting thymoma or non-thymoma were analyzed using descriptive statistics. The results are described in frequencies, percentages, means (standard deviations, SD) or medians (interquartile ranges, IQR) as appropriate. The Wilcoxon Rank-Sum test, Fisher exact and t-test were used to assess the significant differences in the variables between the ThMG and non-ThMG patients with statistical significance set at *p* < 0.05.

The TWA dosages of the MG drugs required for maintaining ADLEs effectively in each half of the follow-up years are shown in medians (IQRs) and compared between the ThMG and non-ThMG patients. The frequency of cases with different post-thymectomy outcomes using the redefined changes in clinical status of the MGFA-PIS classifications for this study were descriptively compared between the ThMG and non-ThMG patients.

### 2.4. Ethics Approval Statement

Approval to carry out this study was granted following the ethical review by the Ethics Committee of the Faculty of Medicine, Prince of Songkla University (EC code: 60-415-14-4, date of approval 17 January 2019). We strictly followed the 1964 Declaration of Helsinki and related amendments as well as good clinical practice guidelines in carrying out clinical research. The analysis was performed in aggregation with all study participants’ personal data fully anonymized to ensure complete protection against disclosing their identities.

### 2.5. Data Availability Statement

All data and analysis methods used in this study are published in this article. There are no data deposited in any pre-print depository sources.

## 3. Results

### 3.1. Study Participants’ Characteristics

There were 85 out of 141 post-thymectomy MG patients eligible for data analysis. The rest were excluded due to no thymic tissue histopathological reports being available or no post-thymectomy follow-ups having taken place in our center. Sixty-eight (80%) of the study cases were female. They comprised 65 non-ThMG and 20 ThMG cases according to thymic tissue histopathology reports. Among the ThMG group, the thymic tissue pathology classifications were type AB in 10 (50%), A in 3 (15%) and B in 7 (35%) (i.e., B1: 2, B2: 4, B3: 1) cases. From seven cases with data of staging available, there were cases with stage 4 (one), 3A (one) and 1A (five) according to French staging system of thymoma.

For the non-ThMG patients, 53 (81.5%) cases were thymic hyperplasia, while 10 (15.4%) and 2 (3.1%) cases were normal and atrophic thymus, respectively. We found a significantly higher age of onset of MG symptoms (44 (41.8, 49.2) vs. 33 (21.5, 44.0), *p* = 0.008) but a shorter time interval from the diagnosis of MG to thymectomy (8.5 (2.0, 12.0) vs. 22.0 (10.2, 26.5), *p* = 0.03) in the ThMG patients. The blood level of thyroid-stimulating hormone (TSH) was found to be significantly lower in the ThMG patients. More than half of the patients in both MG patient groups were classified as at least grade II B by MGFA clinical classification. There were no differences in baseline MGFA classifications and median daily dosages of individual drugs used for treatment of MG before the thymectomies between the two groups (Table 1).

While age at diagnosis of MG (crude OR = 1.07 (1.01, 1.14), *p* = 0.008) and age of diagnosis ≥42 years (crude OR = 6.86 (1.61, 29.23), *p* = 0.005) had significant associations with thymoma by univariable analysis, only male gender was found to have a significant association with thymoma by multivariable analysis (adj OR = 6.62 (0.99, 44.39), *p* = 0.046) when adjusted with age of onset and level of TSH.

### 3.2. Comparison of Time-Weighted Average (TWA) Dosages of Requirements for MG Drugs after Thymectomy between ThMG and Non-ThMG Patients

Twenty-three cases were excluded from analysis of TWAs of MG treatment drugs because the follow-up time was less than one year after the thymectomies. Therefore, 62 MG patients (17 ThMG and 45 non-ThMG cases) were followed for MG clinical improvement with variable lengths of time intervals between the consecutive visits, ranging from 1 to 8 months, depending on their MG responsive symptoms. 

The number of MG patients who continued to have MG treatment drugs after the thymectomies is shown (Table 2). In comparison between the dosage requirements of pyridostigmine, prednisolone and azathioprine between the ThMG and non-ThMG patients for effectively maintaining ADLE during the 5 years (66 months) of follow-ups, the results showed no significant differences in the median TWA dosages of the individual drugs between the two groups in the 6-month intervals of the follow-ups, except for the dosage of pyridostigmine 12 months after thymectomy (Table 3, Table 4 and Table 5 and Figure 1).

### 3.3. Post-Thymectomy Outcomes Assesed by MG-PIS; Frequency of Exacerbations, Crises and Death from MG

After the thymectomies, the patients were followed at the OPD with variable lengths of time intervals, as mentioned earlier. MG-PISs were conducted yearly at the last follow-ups of each year to evaluate the MG status. During the 5-year follow-ups after the thymectomy, most of the patients in both groups were classified as minimal manifestations (MMs) 2–3 according to the MG-PIS, except for a small number of cases in the non-ThMG group, who were assessed as CSR or PR classifications of the MG-PIS, whereas no ThMG patients were assessed as CSR or PR of MG-PIS classifications. We could not assess the statistical differences in post-thymectomy outcomes by MG-PIS between the two patient groups. (Figure 2). The frequency of post-thymectomy exacerbations during the 5-year follow-ups showed no differences between the two MG patient groups (Table 6). Additionally, the overall frequencies of crises were not different between the two patient groups during both the pre- and post-thymectomy periods (Table 7). Nonetheless, the number of both events had decreasing trends after the thymectomies in both groups. No cases died of MG in our study.

## 4. Discussion

Our study found that 20 of 85 (23.5%) of MG patients enrolled in this study were ThMG patients. They had an older age of onset of MG symptoms but a shorter time interval from the diagnosis of MG to thymectomy and a lower blood TSH level compared with non-ThMG patients. There were no differences in the initial MGFA classifications and the dosages of individual MG treatment drugs required before the thymectomies (Table 1). Only male gender had a significant risk of thymoma by multivariable analysis (adj OR = 6.62 (0.99, 44.39), *p* = 0.046) when adjusted with the age of MG onset and level of TSH. We think that it is usual for a neoplasm to present at an older age, and early thymectomy is recommended to be performed once a tumor is found from the investigations. A study from Sweden including 326 MG patients who underwent thymectomy and had thymic tissue histopathology reports reported that thymomas were found in 65 cases (19.9%), thymic hyperplasia in 185 cases (56.7%), and normal thymus in 76 (23.3%) cases. Additionally, MG patients who had thymoma in this study had a higher median age (52 years) of onset of MG symptoms than those who had thymic hyperplasia (22 years), or normal thymus (46 years) [12]. In general, thymic hyperplasia is commonly found in young-onset female MG patients, while thymoma is frequently found in advanced-age-onset males [6,18]. A study by the RARECAREnet Project in Europe during 2000–2007 found that thymoma was more prevalent in males than females with a ratio of 1.7:1, and the annual incidence was nearly double among those aged 65 years or older than those who were under 65 (4.2 vs. 2.3/100,000) [19]. Another 10-year database study reported that 18 of 64 (28.1%) thymoma patients had associated symptoms of MG. There were no differences in age, gender, thymic histopathology class or overall survival between thymoma cases with or without MG. The anti-acetylcholine receptor antibody (AChR Ab), a predominant antibody in MG patients, was more frequently found in the patients who had thymoma and associated MG [20]. AChR Ab was associated in 15–20% of thymoma patients who had clinical symptoms of MG, while 25% of thymoma patients without MG symptoms also had positive AChR Ab results [21]. Therefore, the presence of AChR Ab is not specific for MG. Other antibodies against the post-synaptic sub-receptors besides AChR Ab were muscle-specific kinase (MusK), low-density lipoprotein-receptor-related protein-4 (LRP-4), argin, titin and ryanodine [18]. The different antibodies against the target receptors result in four distinct clinical MG subtypes, i.e., anti-AChR, anti -MusK, anti LRP-4 and triple-seronegative MG [1,2,5,18]. Since AChR Ab tests were not available in our institution during the study time, we could not assess the percentage of associations between AChR Ab and ThMG or non-ThMG patients. For the significantly lower TSH levels in ThMG patients, we had very limited available data from our study to assess. We found one published article reporting milder MG clinical courses in cases with coexisting autoimmune thyroid diseases than those without; however, no differences in risk of MG crises or long-term outcomes were found [11]. We think that based on a similar pathogenesis related to autoimmune disorders, it is not unusual for MG to be associated with autoimmune thyroid diseases.

Since the thymus of an MG patient is the source of post-synaptic receptor autoantibody production, thymectomy is recommended to be performed as early as possible in all generalized MG patients who have no contraindications, especially those who are AChR Ab-positive. This is because thymectomy and long-term and proper-dose immunosuppressive treatment will bring about desirable neurological outcomes [12,13,15]. Recently published studies, including a few systematic reviews and meta-analyses evaluating post-thymectomy outcomes, also supported early thymectomy in non-ThMG patients, because thymectomy resulted in the decreased requirement of MG treatment medicines, the improvement of MG clinical status as evaluated by MG-PIS, or even the complete recovery of MG [6,8,9,10,13,15,16,17,18,22]. Another study advised that extensive transsternal thymectomy should be carried out in generalized, non-Th MG patients aged 18–65 years with a positive AChR Ab test [23], since the delayed removal of an MG patient’s thymus could cause progressive immunological degradation or irreversible damage of the post-synaptic NMJ receptors. Because accumulative evidence suggests that the different clinical subtypes of MG mentioned above are not related to thymic tissue histopathology [1,2,5,18], the removal of the thymus in whatever subtype of MG patients should have satisfactory clinical outcomes. However, the clinical benefits of a thymectomy cannot be expected in a short time after a thymectomy, as such benefits have usually required at least 2 years to manifest after the operation in the existing clinical studies [13,14,24]. It is believed that the delayed clinical benefit is because of the persistence of thymic T cells or circulating antibodies produced by these cells in the serum of MG patients before and a long time after a thymectomy [13,14,21]. For this reason, during the first few post-thymectomy years, despite the extensive surgical removal of the thymus, symptomatic drug, or one or more immunosuppressants are required to control the MG symptoms. Interestingly, a post-thymectomy outcome study suggested that any clinical benefits after a thymectomy should be attributed to the co-treatment with immunosuppressants rather than the operation alone [12]. In conclusion, these available studies strongly confirm the immunological pathogenesis of MG, and therefore, the treatment of MG should be based mainly on an immunomodulation principle in which the early removal of the thymus in clinically indicated patients should be carried out regardless of the type of tissue histopathology. We strongly suggest that thymectomy should be performed in all MG patients who fulfil the indications and who have a low surgical risk. In our practice, we recommended the MG patients who acquired MGFA classification from class II to undergo extensive transsternal thymectomy. The longer time interval from MG diagnosis to thymectomy in our study was partly caused by consent not being obtained from the indicated MG patients due to personal reasons.

We found no significant differences in the daily dosage requirements of pyridostigmine, prednisolone or azathioprine as calculated by the TWAs of the individual drugs between the ThMG and non-ThMG patients. Both MG patient groups required comparable dosages of each MG drug to maintain effective ADLEs through the five-year follow-ups (Table 3, Table 4 and Table 5). In our study, although the thymuses had been extensively removed using the transsternal surgical approach, the clinical benefits as evaluated by the MG-PISs were not obviously different during the follow-up time in each MG patient group. This was possibly because of delays in having the thymectomies carried out, as mentioned previously. However, a trend of decreasing disease severity was suggested by the decreased number of patients assessed as MM 3 of the MG-PIS in both groups over the follow-up period (Figure 2). Additionally, the number of exacerbations and crises showed decreasing trends, although no statistically significant differences between the groups were able to be confirmed (Table 6 and Table 7). No patients died from MG in this study. These findings suggest that thymectomy possibly provides clinical benefits in MG patients regardless of thymic tissue histopathology, but the time expected to achieve the maximal clinical benefits varies. Some studies have reported that an immediate clinical response after a thymectomy was experienced in MG patients who were younger and had milder disease severity (MGFA I-IIA), and when the thymectomies were performed early [9,16,25,26]. One study recommended thymectomy to be performed in MG patients aged under 65 and within one year of disease onset [26]. A literature review also reported that non-ThMG patients aged < 45 were associated with a higher rate of achieving CSR during the follow-ups after a thymectomy [16]. Our MG patients underwent their thymectomies only after an extended time following the diagnosis (Table 1). This was due to the patients expressing worry about the extensive transsternal surgical approach of thymectomy, and thus, consents were usually not given at an early stage when thymectomy was indicated. Otherwise, no video-assisted thoracotomy (VAT), a less invasive thymectomy procedure, was performed in our center. Although VAT was believed to be an equivalent or alternative approach for thymectomy in MG patients in regard with the post-thymectomy outcomes [22], a retrospective study addressed the dual benefits of ‘oncological outcomes’, i.e., thymoma-related overall survival and tumor recurrence, and ‘neurological outcomes’ of MG. The study showed that postoperative MGFA and pathological stage had significant effects on overall survival and tumor-free survival, respectively. For the neurological outcome, the MG patients who had thymoma and underwent extended thymectomy achieved CSR in 34.2% and PR in 38.4%, only 4.1% experienced unchanged or worse outcomes. The less preoperative MG severity evaluated by preoperative MGFA classification had significant association with the achievement of CSR [27].

In this study, we applied the TWAs of the individual drugs used to control MG symptoms to evaluate the post-thymectomy outcomes instead of the MG-PIS, which was used in most previous studies, because we believed that the TWAs of MG drug dosages required would more accurately reflect the patients’ ability to effectively maintain their ADLEs. As the MG-PIS is roughly categorized into five classes of outcomes based on immunosuppressant and/or symptomatic drugs used and the dosage of symptomatic drugs required in the past year, we thought that this would not provide adequate details of the spectrum of post-thymectomy MG symptoms. Moreover, the one-year time for clinical assessment in the MG-PIS was considered as a crude evaluation and did not represent the actual post-thymectomy clinical status well. Since the clinical conditions of MG fluctuate over time, the use of the TWAs of each MG drug required according to the patients’ self-assessment of their ability to satisfactorily achieve ADLE post-thymectomy in our study was likely to be a more accurate assessment. In our study, the dosages of the individual MG treatment drugs were carefully adjusted after patient-and-physician co-assessments based on the information of daily living and working ability provided by the patients during each follow-up visit. Due to the unequal follow-up intervals ranging from 1 to 8 months of time as mentioned, we thought that the recorded MG-PIS assessments during the patients’ visits to the OPD might inaccurately represent the actual post-thymectomy clinical outcomes objectively and longitudinally. In addition, since most of our MG patients worked in the agriculture or labor sectors, satisfaction with their own ability to maintain their ADLEs after the thymectomies followed by some types of MG drugs would be more suitable for evaluating the post-thymectomy outcomes in our setting. Another study of post-thymectomy outcomes also used the TWAs of quantitative myasthenia gravis scores (QMG scores) and alternate-day prednisolone dosages as outcome evaluators to compare between MG patients treated with thymectomy plus alternate-day prednisolone and with alternate-day prednisolone alone [13]. We considered that the QMG score was usually used for evaluating the clinical severity of MG before a treatment intervention, and thus it could not represent the real outcome statuses of post-thymectomy MG cases well. The QMG is also a cross-sectional rather than a longitudinal evaluation score. It is also too complicated to be applied in most OPD service settings where patients with various neurological disorders are serviced, and the spirometer is usually inaccessible in most OPDs, including our OPD service. Hence, the TWAs of individual MG drug dosages required for effectiveness in maintaining post-thymectomy MG patients’ ADLEs were suggested from our study to be used as an alternative evaluation method of clinical response after thymectomy.

## 5. Conclusions

The post-thymectomy clinical benefits on MG symptoms will not be obvious during the first few years. From our study, it is possible that it would take 5 or more years for significant clinical benefits to manifest. Therefore, all post-thymectomy MG patients still require a range of dosages of MG symptomatic and/or immunosuppression medicines to maintain their ability to effectively carry out their activities of daily living or make their livings while awaiting the maximal clinical benefits to occur. Careful adjustment of the MG medicine dosages in response to each post-thymectomy patient’s requirement for carrying out work and daily activities of living satisfactorily is required. An assessment method such as the TWAs of the MG drugs required by the patients as in this study should be appropriate in most cases for the evaluation of post-thymectomy outcomes, because these reflect the real clinical conditions of the patients in the aspect of satisfactory living and working conditions. We confirm from our findings that post-thymectomy clinical improvement is not immediate after thymectomy. We suggest the use of the TWA of the required dosages of the individual MG treatment drugs as a more objective evaluator for post-thymectomy outcome evaluation. The limitation of this study was its retrospective nature and small sample size. Although the MG-PIS is generally accepted for post-thymectomy clinical evaluation, we suggest that clinical evaluation methods which are able to represent post-thymectomy MG patients’ living conditions well should be developed.

## Figures and Tables

**Figure 1 ijerph-20-03039-f001:**
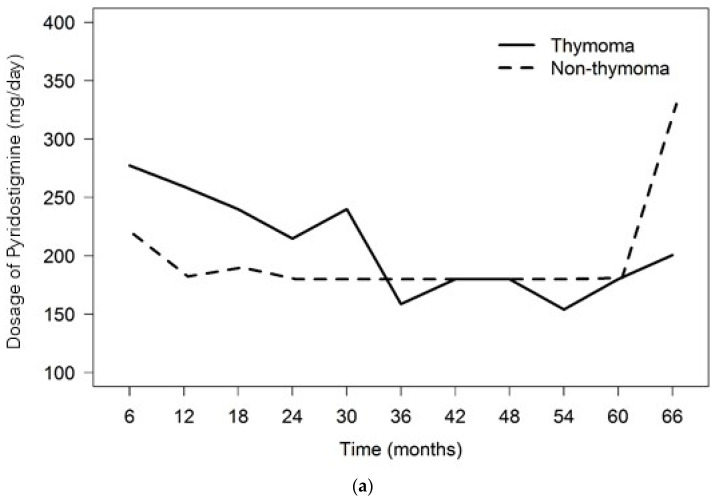
Illustrations demonstrating the time-weighted averages (TWAs) of individual MG treatment drug dosages required during the follow-ups: (**a**) pyridostigmine (Mestinon^R^); (**b**) prednisolone; (**c**) azathioprine.

**Figure 2 ijerph-20-03039-f002:**
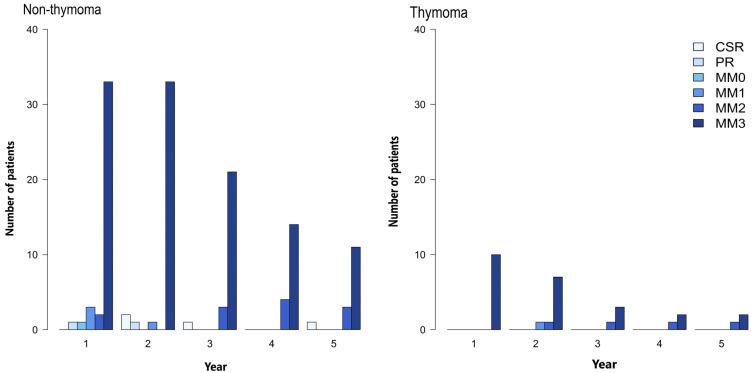
The number of patients distributed in the different levels of post-thymectomy outcomes evaluated by MGFA-PIS in the ThMG and non-ThMG groups during the 5-year follow-ups.

**Table 1 ijerph-20-03039-t001:** Comparison of demographic and clinical characteristics between ThMG and non-ThMG patients before the thymectomies.

Clinical Characteristic	ThMG Patients	Non-ThMG Patients	Total		*p* Value
	(*n* = 20)	(*n* = 65)	*N* = 85		
Sex				Fisher’s exact test	0.214
Male	6 (30)	11 (16.9)	17 (20)		
Female	14 (70)	54 (83.1)	68 (80)		
Age at the diagnosis of MG, Yrs, median (IQR)	44 (41.8, 49.2)	33 (21.5, 44)	36 (25.2, 46)	Rank Sum test	0.008 *
MGFA classification				Fisher’s exact test	0.253
Grade 1	2 (11.1)	16 (27.6)	18 (23.7)		
Grade 2a	5 (27.8)	9 (15.5)	14 (18.4)		
Grade 2b	9 (50)	30 (51.7)	39 (51.3)		
Grade 3a	0 (0)	1 (1.7)	1 (1.3)		
Grade 3b	0 (0)	1 (1.7)	1 (1.3)		
Grade 5	2 (11.1)	1 (1.7)	3 (3.9)		
Time from the diagnosis of MG to thymectomy, months, median (IQR)	8.5 (2, 12)	22 (10.2, 46.5)	12 (7.8, 37.5)	Rank Sum test	0.03 *
TSH µIU/L,	0.8 (0.5, 1.4)	1.5 (1, 2.5)	1.3 (0.8, 2.3)	Rank Sum test	0.017 *
median (IQR)					
Initial dosages of drugs for treating MG					
(a) Prednisolone, mg/d	30 (16.2, 47.5)	20 (11.2, 30)	25 (15, 32.5)	Rank Sum test	0.068
median (IQR)					
(b) Pyridostigmine, mg/d,	240 (180, 300)	240 (180, 300)	240 (180, 300)	Rank Sum test	0.839
median (IQR)					
(c) Azathioprine, mg/d,	50 (50, 100)	87.5 (50, 100)	75 (50, 100)	Rank Sum test	0.878
median (IQR)					

* *p* < 0.05. Abbreviations: MGFA, Myasthenia Gravis Federation of America; TSH, thyroid-stimulating hormone; ThMG, thymoma myasthenia gravis; non-ThMG, non-thymoma myasthenia gravis.

**Table 2 ijerph-20-03039-t002:** Number of the patients who continued to have MG drugs after the thymectomies in each follow-up year between ThMG and non-ThMG patients.

Thymoma (*n* = 17)									Non-Thymoma (*n* = 45)							
	Pyr	p	A	Pyr + p	Pyr + A	p + A	all	None	Pyr	p	A	Pyr + p	Pyr + A	p + A	all	None
Y1	2	1	0	1	2	1	6	0	1	2	1	9	5	0	19	0
Y2	0	0	0	2	2	2	3	0	2	2	0	8	9	0	13	2
Y3	0	0	0	0	0	1	0	0	3	0	0	7	9	0	12	1
Y4	0	0	0	0	0	0	0	0	4	0	0	3	7	0	9	0
Y5	0	0	0	0	0	0	0	0	1	0	0	4	5	0	8	1

Abbreviations: Pyr, pyridostigmine; p, prednisolone; A, azathioprine.

**Table 3 ijerph-20-03039-t003:** Comparison of time-weighted averages (TWAs) of pyridostigmine (Pyr) dosage requirements (mg/d) after thymectomy between ThMG and non-ThMG patients.

Drug and Time (Months) Post-Thymectomy	ThMG *n*= 17 Median (IQR)	Non-ThMG *n* = 45 Median (IQR)	*p* Value
Pyr 6	277.3 (228.2, 390)	218.5 (160.9, 285)	0.055
Pyr 12	259.3 (240, 390)	182.3 (122.2, 242)	0.037 *
Pyr 18	240 (209.6, 285)	190 (120, 240)	0.18
Pyr 24	214.7 (120, 240)	180 (173.9, 240)	0.975
Pyr 30	240 (120, 269.4)	180 (145.5, 238.6)	0.739
Pyr 36	158.7 (120, 211.3)	180 (135, 232.1)	0.53
Pyr 42	180 (120, 180)	180 (150, 240)	0.38
Pyr 48	180 (165, 195)	180 (120, 259.9)	0.878
Pyr 54	154 (126, 210.3)	180 (180, 311.6)	0.464
Pyr 60	180 (150, 330)	181.2 (180, 345)	0.863
Pyr 66	200.7 (152.3, 249)	330 (165, 480)	NA

* *p* < 0.05.

**Table 4 ijerph-20-03039-t004:** Comparison of time-weighted averages (TWAs) of prednisolone (p) dosage requirements (mg/d) after thymectomy between ThMG and non-ThMG patients.

Drug and Time (Months) Post-Thymectomy	ThMG *n* = 17 Median (IQR)	Non-ThMG *n* = 45 Median (IQR)	*p* Value
P 6	17.9 (11, 25.2)	13.2 (8.1, 22.7)	0.446
P 12	9.9 (5, 30.7)	9.8 (5, 17.2)	0.368
P 18	5.8 (2.5, 19.4)	5 (1.4, 13.1)	0.43
P 24	2.5 (0, 5)	5 (1.1, 10)	0.259
P 30	5 (1.2, 9.2)	5 (0, 10)	0.946
P 36	2.7 (1.8, 4.5)	5 (0, 9.4)	0.842
P 42	5 (0, 5)	5 (0, 10)	0.826
P 48	2.5 (0, 17.5)	5 (1.6, 14.6)	0.647
P 54	6.8 (3.8, 12.9)	5 (4.6, 9.5)	0.708
P 60	7.7 (3.9, 16.6)	5 (2.8, 10)	0.708
P 66	10.7 (7.8, 13.5)	0.5 (0.2, 7.8)	NA

**Table 5 ijerph-20-03039-t005:** Comparison of time-weighted averages (TWAs) of azathioprine (A) dosage requirements (mg/d) after thymectomy between ThMG and non-ThMG patients.

Drug and Time (Months) Post-Thymectomy	ThMG *n* = 17 Median (IQR)	Non-Th MG *n* = 45 Median (IQR)	*p* Value
A 6	50 (0, 89.7)	50 (0, 83)	1
A 12	50 (0, 79.8)	56.4 (0, 100)	0.575
A 18	50 (0, 95.1)	50 (27.5, 100)	0.608
A 24	32.6 (0, 50)	50 (3.3, 102.2)	0.092
A 30	36 (0, 55.5)	50 (0, 114.9)	0.384
A 36	37.5 (6.2, 64.9)	50 (0, 125)	0.387
A 42	25 (0, 50)	50 (11.6, 102.3)	0.199
A 48	25 (0, 51.5)	98.3 (13.6, 119.2)	0.172
A 54	41.1 (24.2, 56)	50 (0, 135.2)	0.535
A 60	50 (50, 82.5)	55 (12.5, 143.8)	0.931
A 66	38.9 (33.3, 44.4)	103.3 (89.2, 126.7)	NA

**Table 6 ijerph-20-03039-t006:** Comparison of the frequencies of exacerbations after thymectomy between ThMG and non-ThMG patients during the 5-year follow-ups.

Exacerbation	ThMG	Non-ThMG	Total		*p* Value
Year 1	10	40	50	Fisher’s exact test	0.671
No	7 (70)	32 (80)	39 (78)		
Yes	3 (30)	8 (20)	11 (22)		
Year 2	9	37	46	Fisher’s exact test	1
No	8 (88.9)	32 (86.5)	40 (87)		
Yes	1 (11.1)	5 (13.5)	6 (13)		
Year 3	7	26	33	Fisher’s exact test	0.145
No	4 (57.1)	22 (84.6)	26 (78.8)		
Yes	3 (42.9)	4 (15.4)	7 (21.2)		
Year 4	4	19	23	Fisher’s exact test	1
No	4 (100)	17 (89.5)	21 (91.3)		
Yes	0 (0)	2 (10.5)	2 (8.7)		
Year 5	4	15	19	Fisher’s exact test	0.178
No	2 (50)	13 (86.7)	15 (78.9)		
Yes	2 (50)	2 (13.3)	4 (21.1)		

**Table 7 ijerph-20-03039-t007:** Comparison of the frequencies of MG crises pre- and post-thymectomy between ThMG and non-ThMG patients.

Pre-Thymectomy Crisis	ThMG *n* = 14	Non-ThMG *n* = 48	Total	Test Stat	*p* Value
No	9 (64.3)	34 (70.8)	43 (69.4)	Fisher’s exact	0.744
Yes	5 (35.7)	14 (29.2)	19 (30.6)		
Post-thymectomy crisis					
No	12 (85.7)	44(91.7)	56 (90.3)	Fisher’s exact	0.610
Yes	2 (14.3)	4 (8.3)	6 (9.7)		

## Data Availability

The study data and analysis methods were described in the article. No data were deposited in any preprint servers.

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
