# Peer review of "Comparison of Short-Term Post-Thymectomy Outcomes by Time-Weighted Dosages of Drug Requirements between Thymoma and Non-Thymoma Myasthenia Gravis Patients"

_ijerph, 2023, doi:10.3390/ijerph20043039_

Round 1

Reviewer 1 Report

Dear Editor and Authors

 I had the pleasure to review this interesting article entiled “Comparison of short-term post-thymectomy outcomes by time-2 weighted dosages of drug requirements between thymoma and 3 non-thymoma myasthenia gravis patient”, by Wiboonpong and colleagues.

Authors evaluated the neurological outcomes of patients underwent transsternal thymectomy by an accurate surveillance period.

The article in overall well-structured and interesting but there are some issues and concern that should be solved.

-          Limitation and strengths should be highlighted in the whole text. The study has different limitations, but I agree with authors on the use of the TWA of the required dosages as a more objective evaluator for post-thymectomy outcome evaluation. I would focus the whole text on this aspect.

-          40% of data missing is a high rate, maybe it could be better adding some incomplete data to get stronger results.

-          WHO classification of thymoma should be reported properly (B1,B2 etc.)

-          Unit of measurement should always be reported (years, months etc)

-          Indication for thymectomy is well established world-wide, could the author explain the delayed surgery in case of tumor (8.5 months) as well as in case of non-thymomatous MG (22 months)

-          Since anti-AChR e anti -MusK, are not available, on which bases authors decided to address nonTh MG patients to surgery. This part should be more extensively discussed. Moreover, why authors evaluated TSH in all patients.

-          In daily practice, many patients do not assume pyridostigmine due to the collateral effect as well as Azathioprine is administered in few patients. Could authors add a table in which are reported the number of patients assuming these therapies.

-          Drug dosage is often related to the body mass index or creatine clearance; have authors evaluated this correlation?  

-          The most dangerous period is the perioperative time. Have authors data on eventual post-operative crisis?

-          -Steroid have been demonstrated an oncological role in reduce the tumor mass and prevent post-operative crisis. Several centers improve the perioperative dosage of steroid with this aim. What was he policy of authors?

-          Could authors provide data on the pathological thymoma status as stage and radicality

-          Discussion might benefit adding this article (PMID: 33358889)

-          Please check any typos (i.e. line 134, 267-270)

Author Response

Dear Reviewer 1

Please see the attached file of response to comments below

Regards,

Pornchai Sathirapanya

Reviewer 2 Report

Thank you very much for sending me an interesting article entitled "Comparison of short-term post-thymectomy outcomes by time-weighted dosages of drug requirements between thymoma and non-thymoma myasthenia gravis patients" for review.

In their paper, the authors present the long-term results of the surgical treatment of myasthenia gravis, and compare the outcomes between the groups of patients with and without thymoma.

The article is interesting and is written in good quality English. The individual parts of the main text are written according to the guidelines and contain all the necessary information. The introduction provides a good background for research. The methods used are well described and appropriate. The results are presented in a clear and detailed way. Tables and charts are complementary to the text, the authors have avoided repeating information already contained in the text. In the introduction and in the discussion, the authors refer to the current literature.

Overall, I found the article interesting and well written. I have no comments to the article and suggest its publication.

Author Response

Dear reviewer 2

Please find the file of response to comments below.

Regards,

Pornchai Sathirapanya

Reviewer 3 Report

­­­­­­­­­­­­­­­­­­­­­­­This is a good study that compares the clinical characteristics of thymoma myasthenia gravis and non-thymoma myasthenia gravis patients while comparing the five-year outcomes post thymectomy, frequencies of exacerbations and deaths due to Myasthenia gravis.

Strengths:

- The study details the trend of clinical improvement during the first 5 consecutive years after thymectomy in thymoma myasthenia gravis and non-thymoma myasthenia gravis patients.

- The references are most recent and comprehensive.   
- Interpretation and presentation of previous studies is accurate.             
- No major suggestions for improvements.         
- Clarity and context in this paper are good.

Minor comments:          
- Introduction could include more background information.

- The figure 2 could use a better/larger textual font for the axes as they are currently too small.

- Language editing and proper punctuation is needed.   
 For example: Line 353 needs spacing to be corrected.   
In line 215 did the authors mean “20 of 85 (23.5) MG patients”?

Author Response

Dear reviewer 3

Please the attached file of response to comments below.

Regards,

Pornchai Sathirapanya
